# Evaluating the measurement properties of the Hindi Integrated Palliative Care Outcome Scale (IPOS) in advanced cancer patients receiving home-based palliative care in India

Tushti Bhardwaj[1,2]*, Rachel L. Chambers[1], Harry Watson[1], Ambika Rajvanshi[3], Reena Sharma[3], Irene J. Higginson[1,4], Mevhibe B. Hocaoglu[1]*

1 Cicely Saunders Institute of Palliative Care, Policy and Rehabilitation, King's College London, London, United Kingdom, 2 Department of Social Work, Dr. Bhim Rao Ambedkar College, University of Delhi, Delhi, India, 3 CanSupport, Delhi, India, 4 King's College Hospital NHS Foundation Trust, London, United Kingdom

* Mevhibe.hocaoglu@kcl.ac.uk (MBH); tushti.1.bhardwaj@kcl.ac.uk, tushti.bhardwaj@bramb.du.ac.in (TB)

## Abstract

A culturally appropriate person-centred approach is needed in Indian palliative care settings to address the holistic concerns most distressing for patients with cancer. The study aimed to evaluate the measurement properties of the Hindi Integrated Palliative Outcome Scale (IPOS), a person-centred tool to assess the physical, emotional, and care-related concerns of patients living with serious illness in resource-limited settings. This validation study was conducted with 240 adults living with cancer and receiving home-based palliative care services through CanSupport, a charitable organisation operating across Delhi and National Capital Region. Using consecutive sampling, participants completed the Hindi IPOS, EuroQol 5-Dimension 5-Level questionnaire (EQ-5D-5L), and demographic schedules through in-person interviews. Half of the participants completed two follow-up assessments at four to six week intervals. Participant recruitment and outcome assessments at baseline and the first follow-up were conducted in person by home-based palliative care teams. Due to COVID-19 constraints, the second follow-up assessment for a small subset of participants (10 patients) was completed via video call. Measurement properties assessed included structural validity (exploratory and confirmatory factor analysis), internal consistency, test–retest reliability, construct validity, and responsiveness to change. Confirmatory factor analysis supported a three-factor structure (CFI = 0.845, RMSEA = 0.062, TLI = 0.794), aligning with the hypothesized model. Internal consistency was moderate to high for physical ($\alpha = 0.67$) and emotional ($\alpha = 0.74$) subscales, while the communication and quality of care subscale demonstrated low consistency ($\alpha = 0.35$). Construct and known-groups validity were supported by associations with EQ-5D-5L and cancer stage (p < 0.001). The physical subscale detected meaningful changes over time. Open-text responses provided insight into unaddressed

**Data availability statement:** The minimal data set underlying the results presented in this manuscript—including the values behind summary statistics, measures of variation, and data used to generate figures and tables—is available from the Palliative Outcome Scale (POS) Team upon reasonable request. Due to ethical and confidentiality restrictions protecting the privacy of participants, the full dataset cannot be made publicly available. The Institutional Ethics Committee – CanSupport and the collaborating academic institution, Dr. Bhim Rao Ambedkar College, University of Delhi, have imposed restrictions on public data sharing because certain demographic variables may allow for participant re-identification even after de-identification procedures. Researchers who meet the criteria for access to confidential data may submit a formal request to the Palliative Outcome Scale (POS) Team, who serve as the non-author institutional point of contact for data access inquiries, at palliativecare@pos-pal.org. Requests will be reviewed in accordance with institutional and ethical guidelines. Ethics approval for this study was obtained from the Institutional Ethics Committee – CanSupport.

**Funding:** The author(s) disclosed receipt of the following financial support for the research, authorship, and/or publication of this article: Support was from the National Institute for Health Research (NIHR), Applied Research Collaboration, South London, hosted at King's College Hospital NHS Foundation Trust, Palliative Outcome Scale Development Team and Cicely Saunders International (Registered Charity No. 1087195). The funders of the study had no role in study design, data collection, data analysis, data interpretation, or writing of the report.

**Competing interests:** The authors (TB, RLC, HW, AR, RS, IJH, MBH) declare that there are no potential conflicts of interest with respect to the research, authorship, and/or publication of this article. IJH, RLC, HW, and MBH are members of the POS Development Team at KCL.

symptoms and psychosocial needs. IPOS Hindi is a valid, reliable, and culturally appropriate tool to assess palliative care needs in patients with advanced cancer. It supports person-centred care and can inform policy and practice of palliative care delivery.

## Introduction

The global burden of serious health-related suffering (SHS) has risen sharply, with over 73 million people affected in 2021—an increase of 74% since 1990 [1]. Most of this burden is concentrated in low- and middle-income countries (LMICs), where access to timely, quality palliative care remains limited. The Lancet Commission calls for integrating palliative care into universal health coverage to address this growing need [1]. The *Global Atlas of Palliative Care* highlights the importance of culturally appropriate approaches, particularly in resource-limited settings [2]. Palliative care is holistic and person-centred, addressing not just physical symptoms but also emotional, social, and spiritual needs [3]. There remains a critical lack of robust tools to effectively monitor progress in alleviating pain and other forms of serious health-related suffering. Improved metrics and stronger evidence are urgently needed to support priority setting, accurately assess the global need for palliative care, and guide countries in implementing effective policies and programs [4].

Palliative care services in India were introduced approximately forty years ago [5]. In 1984, palliative care provisions were introduced in the National Cancer Control Programs, making pain relief available in the community. In the absence of a national palliative care policy, low coverage and unequal access to palliative care are ongoing problems in India [6]. Only 2% of the population have access to palliative care services [6–8]. Home-based community palliative care models used in India are mainly provided by charity organizations or hospital home care teams. Home-based palliative care is delivered by a multidisciplinary team consisting of a medical doctor, a nurse, and a counsellor.

Home based model adopted by the charity organization where this research was implemented follows the symptom severity design to offer services and decide the frequency of visits. Patients with severe pain, breathlessness, persistent vomiting, or other uncontrolled symptoms are visited up to five times a week and designated as high priority needs patients. These patients are often on morphine, bedridden, and can have fungating wounds. More stable patients whose symptoms are well controlled with oral medicines are designated as medium priority patients and visited every 10–15 days. Patients who are symptom-free for 2–3 years are designated as low priority and visited once a month in person or followed up by phone if they have moved out of area.

In a resource limited setting, routine assessments of patient-centred outcomes are needed to understand the needs and concerns of patients [9–11] to appropriately provide specialized palliative care services to those who really need it most. Integrated Palliative Outcome scale (IPOS) is widely used patient-centric measure built on a reflective model capturing concerns and needs of patients living

with life-limiting, advanced illness in community and hospital care settings [12]. There is an urgent need for a culturally adapted and validated brief palliative care measure to inform practitioners where the limited resources are most urgently needed. Previously, we have translated and culturally adapted IPOS to accurately describe the most important concerns of patients living with cancer in India [13]. In this study we field test and evaluate the measurement properties of the IPOS in Hindi-speaking adult cancer patients receiving home palliative care services in Delhi National Capital Region (NCR), India. The findings of this study have implications for cross-cultural palliative care research and practice [14], and benefit Hindi-speaking patients with cancer across India and other parts of the world.

## Methods

Ethical approval was obtained from the Institutional Ethics Committee for Human Research of CanSupport Charity (IEC-CanSupport-1/2021 dated 5th Feb 2021). The original developer of IPOS provided permission for cultural adaptation and validation of the measurement instrument.

### Inclusivity in global research

Additional information regarding the ethical, cultural, and scientific considerations specific to inclusivity in global research is included in the Supporting Information file (S1 Checklist).

### Design

This is a validation study, including repeated measures to evaluate test–retest reliability and responsiveness over time.

### Setting

Patients were recruited from CanSupport Charity which provides free home-based palliative care support to persons with life-limiting illnesses in all regions of Delhi NCR, India [15]. Patients to the Charity were referred by the treatment hospitals across the city. Referrals to the service were made based on symptoms burden, functional decline, or psychosocial need across all cancer stages. Therefore, some patients with earlier-stage disease were part of the service, hence included in the sample.

### Participants

Adult (≥18) cancer patients receiving home-based palliative care from CanSupport, who were able to communicate and had the capacity to consent, scheduled to be visited by the home-based care team during the planned data collection period were eligible to participate in this study. Initial contact with participants was made by telephone to provide study information and obtain verbal agreement to be approached. Written informed consent was subsequently obtained in person prior to data collection. Participant recruitment and the first two assessments were conducted in person by the home-based palliative care teams during routine clinical visits. Only 10 patients during second follow up could not be reached in person due to covid restrictions in their respective geographical regions and hence contacted through virtual video calls. A sample of 240 participants were recruited using consecutive sampling techniques. The sample size was decided following recommendations by the IPOS developers [16] ensuring a minimum of 10 participants per questionnaire item [17] setting a minimum requirement of 170 patients. During the study recruitment period, (February-March 2021) approximately 400 patients received services from the home-based palliative care services of the charity organization in Dehi NCR. All eligible patients during the recruitment period were given opportunity to participate. Out of these, 240 consented and were enrolled in the study. Patients who were not adults, too sick to participate, unable to speak and unwilling to consent were excluded. To ensure representation of the patients from all parts of Delhi NCR, 24 teams from varied geographical locations of Delhi NCR collected data and recruited ten patients each making a sample of 240 patients.

## Data collection

All participants completed IPOS at baseline, and half of them completed it at two more time points over a period of three months. Baseline data collection took place between February - March 2021, the first follow-up between March - April 2021 and the second follow-up April to May 2021. At baseline, IPOS Hindi, EQ-5D-5L Hindi versions and a demographic sheet were administered. At follow-up visits, the participants responded to a Global Rating of Change Question and IPOS Hindi.

Counsellors administered the hardcopies of the study packs through in-person interview with the patient. The Principal Investigator of the study (TB) trained the counsellors on data collection, confidentiality, coding procedure, and monitored the data collection process, addressed any questions through telephone follow-ups with the counsellors. Training was repeated to ensure quality at each data collection time-point. At follow-up, the same counsellor administered the tool pack to the same patient. The completed forms were returned to the PI immediately following visits. The PI transferred data from paper to electronic format.

## Measures

1. Integrated Palliative Outcome Scale (IPOS) is widely accepted among patients with advanced diseases [12], is responsive to change, and used in clinical care and research for cancer and non-cancer patients [12,18]. In addition to 17 closed ended questions on how specific concerns/problems affect them (0 = not at all to 4 = overwhelmingly), it includes an open-ended question, 'What have been your main problems or concerns over the past week?' which generates textual data. The original English is available to download from IPOS website [19] and the Hindi language version is attached in the supplementary files (Fig A in S1 File).

2. *EQ-5D-5L* is a generic measure of health status developed by the EuroQol Group and covers five items around domains, namely mobility, self-care, usual activities, pain/discomfort, and anxiety/depression. [20] A vertical visual analogue scale (VAS) where the endpoints are labelled as 'The best health you can imagine' and 'The worst health you can imagine' captures the patient's perception of their health status

3. The Global Rating of Change Question (GRCQ) were administered first before IPOS Hindi in the follow-up assessments. The GRCQ asked, "Compared to the last time we visited you, how do you feel today?' to which participants were asked to respond as 'Better', 'Same' or 'Worse'.

Relevant clinical and demographic data were also collected for all respondents during the first visit. Demographic details related to family and socio-economic status were assessed as self-perceived variables, while clinical data were collected from medical records. Data on co-morbidities were reported by the participants themselves. Cancer stage was recorded based on hospital documentation at the time of referral. The home-based palliative care service or research team did not independently assess disease stage.

## Data management and quality assurance

Data was transferred from paper to electronic format by a team of researchers led by TB. IBM SPSS version 28 and SPSS AMOS were used for data management and analysis. MH drafted the SPSS spreadsheet in consultation with TB through MS Teams video meetings. Cancer diagnoses were re-coded using ICD-10 codes. To ensure the quality of data entry and controlling chance of human error, patients were assigned a number from 1 to 240 as their unique ID codes. Twelve questionnaire packs (5% of the total sample) were randomly selected by the PI (TB) through randomly generated numbers between 1–240 by MH. The corresponding questionnaire packs and responses were compared with electronically entered data during live online meetings with TB, MH, RC and HW to check the accuracy of data entry.

## Data analysis

**Qualitative analysis.** The open-text questions on the three main concerns (IPOS Q.1) were thematically grouped, counted, and described using a word cloud. RC and HW identified the themes, and all authors reviewed them. The size of words in the word cloud represented the relative frequency of occurrence of the symptom. The more participants reported a concern, the bigger the size of the word.

**Measurement properties.** For this study, we evaluated the psychometric qualities of the Hindi IPOS in line with internationally accepted recommendations, including COSMIN criteria [21] and guidance from the US FDA on Patient-Reported outcomes [22] (S2 File). Pairwise deletion [23] method was used for the missing data, where all available data is used for analysis. For IPOS, the total scores and the sub-scale scores have been calculated by simple addition of relevant item scores. No weighting has been used. Missing values have not been imputed, and cases with missing values were not included in analysis. If a participant was missing an item score from a subscale, the subscale score was not calculated for them.

Structural Validity: To assess the scale's structural validity and acceptability, we first explored the subscales with Exploratory Factor Analysis (EFA), principal axis extraction and oblique method (direct oblimin) [24] and compared them to the 3-factor structure proposed within the original measure [12]. Secondly, we undertook a first-order Confirmatory Factor Analysis (CFA) to assess how well the hypothesized 3-factor model fits the observed data. The model parameters were estimated using the Full Information Maximum Likelihood (FIML) method to make maximal use of all data [25]. Post-hoc modifications were undertaken to improve model fit. We evaluated model fit for standardized estimates of the model using chi-square, ratio of chi-square and degrees of freedom, confirmatory fit index (CFI), Tucker-Lewis index (TLI) and root mean square error of approximation (RMSEA) for small samples [26,27].

Thirdly, once the structural validity was established, IPOS Hindi acceptability among the target group was assessed by examining the distribution of items and subscale scores, floor and ceiling effects, and data completeness with Missing Value Analysis.

Reliability: We evaluated reliability in several ways. We examined the internal consistency and determined agreement among responses to items. Internal consistency of each subscale was examined using Cronbach's alpha, which reflects how closely related the items are. [28] In addition, we assessed item–total correlations, setting 0.30 as the minimum acceptable threshold for an item to adequately differentiate between patients with more or less severe symptoms. [29] An item-total correlation below 0.30 implies that the item cannot discriminate well between patients severely and less severely affected by cancer [30,31].

We also examined test-retest reliability in patients who felt the same since the last time they completed IPOS Hindi at the follow-up assessments. To the best of our knowledge, the evaluations took place under similar care conditions.

We anticipated that test–retest analysis would demonstrate stability over time for patients whose health status remained unchanged. Specifically, we expected paired t-tests comparing baseline and follow-up scores to reveal no statistically significant differences, consistent with the assumption that mean change scores in such cases should approximate zero.

We assessed inter-rater reproducibility to understand the consistency with multiple raters administering IPOS Hindi. Interclass Correlation Coefficients (ICCs) using two-way mixed effect models were obtained and ICCs for average measures were reported. The choice of the model was informed by the fact that multiple administrators from a selected pool carried out the assessments and consistency is critical [32]. ICCs less than 0.5 are poor, between 0.50 and 0.75 are moderate, between 0.75 and 0.90 are good, and above 0.90 indicates excellent reliability [32]. To evaluate consistency across different assessors, intraclass correlation coefficients (ICCs) were calculated using a two-way model. Using this model, both the raters and the subjects are considered sources of random effects. This model is often used to generalize the findings to any raters, beyond those used in this study [32,33].

We also estimated the Standard Error of Measurement (SEM) to quantify potential error in repeated assessments, with values between 0.8 and 0.9 generally regarded as an indication of satisfactory precision [34].The formula for calculating Standard Error of Measurement (SEM) is given below:

$$SEM = Standard\ Deviation\sqrt{1 - Reliability}$$

SEM between 0.80-0.90 is considered evidence of adequate measurement precision [35].

Construct Validity: We assessed the construct validity in several ways. First, we examined the associations between IPOS Hindi with EQ-5D-5L individual items as well as ratings of health status and cancer staging (early and advanced) according to 'a priori' hypotheses. We hypothesized that patients who have more problems (higher scores) with aspects of quality of life captured by EQ-5D-5L and have worse self-rated health status indicated by lower scores on the VAS scale, are more severely affected by their condition and would therefore have higher IPOS Hindi scores. We formed two groups based on cancer stage at baseline, categorizing patients into 'early', and 'advanced' groups. We examined divergent validity by hypothesizing that patients at the advanced stages of cancer (Stage III and IV) would have higher IPOS Hindi scores, compared to those at earlier stages (Stage I and II). The strength and direction of the association of IPOS Hindi with EQ-5D-5L, VAS Scale, and with stages of cancer were evaluated using Spearman rank-order correlation coefficient. Cancer staging is based on pathological classification and extracted from medical notes by the data collection team.

We examined discriminative or known-groups validity within these subgroups using independent sample t-tests. We hypothesized that the group of patients with advanced cancer have statistically significant ($p \leq 0.05$) higher mean IPOS Hindi scores compared to those with early cancer. For interpretation of responsiveness, we classified effect sizes (Cohen's d point estimate) as follows: values above 0.8 indicated a strong effect, those between 0.5 and 0.8 were viewed as moderate, scores from 0.2 to 0.4 as modest or small, and anything below 0.2 as having little to no meaningful impact [36,37].

Responsiveness: We examined the responsiveness, the ability of IPOS Hindi to detect a change in the patient's status [38] to inform sample size decisions evaluating effectiveness in future trials. We hypothesized that IPOS Hindi scores would capture improvement or deterioration and would stay the same when there were no changes in a patient's status. We first calculated differences in IPOS Hindi scores between (i) baseline and time 1 (T1), (ii) time 1 (T1) and time 2 (T2) and (iii) baseline and time 2 (T2). We subtracted the earlier score from that of the later assessment. A positive change score indicated deterioration. We used patient self-rated anchors to define clinical change, based on the patient's response to the global rate of change questions asked at the beginning of each follow-up assessment. We calculated effect sizes to capture clinically important changes [39]. Based on the self-rated anchor, the patients' response to the Global Rating of Change question, patients who showed improvement and responded that they were 'better' compared to the previous assessment were included in effect size calculations [37,40].

The effect sizes were calculated using the formulas below for baseline (T0) and first follow-up (T1) assessment, T1 and Second follow-up (T2) assessments respectively for each of the subscale scores [39]:

$$ES = \frac{Mean\ Change\ Score\ of\ Subscale\ (T1 - T0)_{improved}}{Standard\ Deviation\ of\ T0_{same\ (unchanged)}}$$

$$ES = \frac{Mean\ Change\ Score\ of\ Subscale\ (T2 - T1)_{improved}}{Standard\ Deviation\ of\ T1_{same(unchanged)}}$$

To capture the ability of IPOS Hindi to detect change in general, Standardised Responsive Mean (SRM) and the effect sizes for change from baseline to final assessment were calculated using the formula below:

$$SRM = \frac{(Mean\ Change\ Subscale\ Score\ from\ baseline\ to\ second\ follow-up)_{total\ group}}{(SD\ of\ Mean\ Change\ Subscale\ Score\ from\ baseline\ to\ second\ follow-up)_{total\ group}}$$

We calculated Minimally Important Change (MIC) based on median change scores in patients who had *improved* or *deteriorated* between baseline and first follow-up assessments.

## Results

### Quality assurance

During accuracy checks of data entry, a total of 13 errors were detected in 12 records of the patients from a total of 143 variables, giving us a very small and acceptable error rate of 0.7%, requiring no further checks.

### Participant characteristics

The median age of the participants was 51 years, 69.6% were women, 27.6% had no formal education, 80% were Hindu. Breast cancer was the most prevalent cancer (28.2%), most were living with Stage 2–4 cancer. Hypertension and diabetes were the most common morbidity, majority were living with at least one self-reported co-morbidity. A big majority (83.75%) were either poor or belonged to lower middle class and were struggling with their current income (62.4%), (Table 1).

### Qualitative entries on Main Concerns of the patients

IPOS helped us to understand the main concerns of the participants. A total of 105 patients reported three main concerns. Pain was the most reported concern, followed by fatigue, financial difficulty, being worried/tense, issues with wound or tube care, poor appetite, constipation and issues with their mouth like dryness, sore mouth, difficulty in eating food or mouth opening. Participants' worry was mainly related to the health of other family members, settling down of children in life and how they could not be able to organize important life events such as weddings for their children. Fig 1 summarizes the data in a word cloud.

### Measurement properties

Data completeness and structural validation.

Data is complete, with acceptable levels of missingness. Twelve out of 17 items have 15% or more floor effects, and 4 items ceiling effects, but the subscales have no floor or ceiling effects (Table 2).

At the subscale level, floor or ceiling effects are within acceptable levels (Table 2).

The 3-factor solution was very close to the original IPOS [12] (Table A in S1 File). The confirmatory factor analysis (CFA) fit indices (CFI = 0.719 and RMSEA = 0.081, TLI = 0.638) indicated medium fit of the model to the data ($\chi2 = 307.252$, df = 119, $\chi2/df = 2.582$, $p < 0.0001$) (Fig B in S1 File). Post-hoc modification involved parceling the three variables from physical domain, namely nausea, vomiting and constipation, to make them one gastrointestinal variable. The model fit following post-hoc modifications was improved with CFI = 0.845 and RMESA = 0.062, TLI = 0.794 ($\chi2 = 171.473$, df = 90, $\chi2/df = 1.905$, $p < 0.0001$). Since the CFI and RMSEA parameters approached the minimums, they were within the required defined parameters recommended for small samples [41]. Thus, the CFA indicates that there is a relatively good fit between the hypothesized model and the observed data confirming cross-cultural validity of IPOS.

The standardized parameter estimates of the modified model are shown in Fig 2. Physical factors accounted for 70–17-%- of variance, Emotional factor for 75–52% and Communication (Communication and Quality of Care) show large variation and account for 82–13% of the relevant IPOS items. Communication subscale had poorly loaded items where information seeking and practical problems had factor loadings of 0.30 and 0.13 respectively (see Fig 2 for distribution of scores), while in Turkish IPOS [9] emotional concerns had poor loading items which behaved quite well in IPOS Hindi. The physical and emotional factors have moderate correlation (0.59), emotional and communication have weak correlation (0.36), and physical and communication factors have poor correlation (0.08).

**Table 1. Demographic and clinical characteristics of participants (n = 240).**

| Variable | Participants | |
|---|---|---|
| | **N** | **% *** |
| **Age (years)** | | |
| Mean (± SD) | 51.7 (± 13.1) | – |
| Median | 51 | – |
| Range | 20 - 90 | – |
| **Sex** | | |
| Female | 167 | 69.6 |
| Male | 73 | 30.4 |
| **Marital status** | | |
| Single | 13 | 5.4 |
| Married/having partner | 184 | 76.7 |
| Widowed | 37 | 15.4 |
| Divorced | 6 | 2.5 |
| **Education**** | | |
| No formal education | 66 | 27.6 |
| Primary (Class V) | 56 | 23.4 |
| Secondary (Class X) | 59 | 24.7 |
| Senior secondary (Class XII) | 22 | 9.2 |
| Bachelor's degree | 26 | 10.9 |
| Master's degree | 10 | 4.2 |
| **Social economic status**** | | |
| Poor | 94 | 39.3 |
| Lower Middle | 107 | 44.8 |
| Middle | 27 | 11.3 |
| Upper Middle and High | 11 | 4.6 |
| **Income status** | | |
| Living comfortably on present income | 30 | 12.8 |
| Coping on present income | 58 | 24.8 |
| Difficult on present income | 89 | 38.0 |
| Very difficult on present income | 57 | 24.4 |
| Don't know or prefer not to say | <5 | – |
| **Family structure** | | |
| Nuclear (married couple living with children) | 144 | 60.0 |
| Joint/Extended (number of married couples living with their children and parents) | 96 | 40.0 |
| **Household size** | | |
| 2–4 people | 104 | 44.1 |
| 5–6 people | 88 | 37.3 |
| More than 7 people | 44 | 18.6 |
| Alone | <5 | – |
| **Accommodation** | | |
| Government or company owned | 11 | 4.7 |
| Their own | 161 | 68.8 |
| Rented | 62 | 26.5 |
| Charitable shelter | <5 | – |

*(Continued)*

**Table 1.** (Continued)

| Variable | Participants | |
|---|---|---|
| | **N** | **% *** |
| **Living in a slum (Yes)** | 21 | 8.8 |
| **Religion** | | |
| Hindu | 192 | 80.0 |
| Other (Muslim, Christian, Sikh etc.) | 48 | 20.0 |
| **Cancer type**** | | |
| Breast (excludes skin of breast) | 67 | 28.2 |
| Lip, oral cavity, and pharynx | 55 | 23.1 |
| Female genital organs | 39 | 16.4 |
| Digestive organs | 31 | 12.9 |
| Respiratory system and Intrathoracic organs | 13 | 5.4 |
| **Cancer stage** | | |
| Stage 1 | 12 | 5.0 |
| Stage 2 | 66 | 27.7 |
| Stage 3 | 59 | 24.8 |
| Stage 4 | 67 | 28.2 |
| Unknown | 29 | 12.2 |
| Not applicable | 5 | 2.1 |
| **Comorbidities** | | |
| Hypertension | 46 | 19.2 |
| Diabetes | 27 | 11.3 |
| Other (Congestive cardiac failure, Chronic pulmonary disease, Peripheral vascular disease, Renal disease, Peptic ulcer disease, Depression, Hemiplegia, liver disease, connective tissue disease) | 31 | 12.9 |
| **Number of comorbidities** | | |
| 1 | 166 | 69.2 |
| 2 | 49 | 20.4 |
| 3 | 20 | 8.3 |
| 4 | 5 | 2.1 |
| Mean (± SD) | 1.43 (± 0.7) | – |
| Median | 1 | – |
| **History of cancer in the family (Yes)** | 29 | 12.1 |

*Percentages are based on available data.

** Based on Indian Schooling System

***Self perceived socio-economic status

**** Five most prevalent cancers are presented.

Reliability: Physical and Emotional subscales show high to moderate internal consistency reliability (respectively α = 0.67 and α = 0.74). In contrast, the communication and quality of care subscale have low internal consistency (α = 0.35).

Items of the communication and support subscale and 3 items of the physical subscale (poor appetite, sore/dry mouth and drowsiness)do not discriminate well between patients severely and less severely affected by cancer [30] (Table 3).

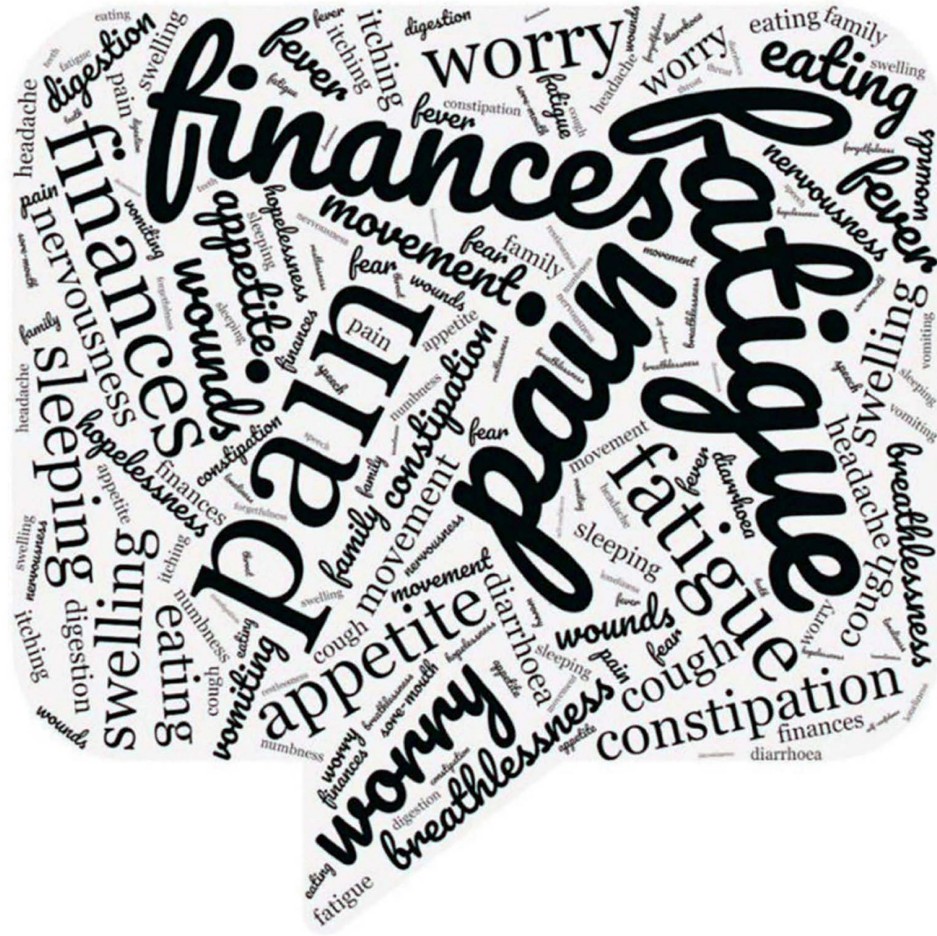

**Fig 1. Word cloud of the main concerns (the size of the script representing frequency).**

Test-retest reliability analysis failed to reject the null hypothesis ($H_0$), and there was no significant difference between baseline and follow-up, confirming that IPOS could appropriately capture the concerns of the patients. (tested here is that the mean difference of IPOS subscale scores among baseline and follow-up assessment subscale scores in patients who felt the same) (Table B in S1 File). Interrater reliability is good with an ICC of 0.76 (0.71-0.80 95%CI, n = 209) for average measures. Standard Error of Measurement (SEM) is low (SEM = 4.19).

*Validity:* As hypothesized, the associations between IPOS Hindi with EQ-5D-5L individual items and rating of health status were all significant ($p < 0.001$) (Table C in S1 File). The correlations between IPOS Hindi subscales and EQ-5D-5L items and VAS scores are in the direction predicted, positive with EQ-5D-5L items as higher scores on both indicate worse outcomes, and negative with VAS as higher scores indicate better health, and moderate levels. For example, IPOS Hindi Physical subscale correlates moderately with mobility, usual activity, and anxiety, but weakly with self-care, pain/discomfort, and self-rated health. Interestingly, emotional items seem to have moderate correlations with mobility, self-care and usual activity, and weak correlations with pain/discomfort, anxiety, and self-rated health.

IPOS Physical subscale has discriminative or known-groups validity where patients with advanced cancer have higher mean IPOS Hindi scores (M = 13.2, SD = 5.65) compared to those with early cancer (M = 10.4, SD = 4.93), t (163.6) =-3.5, $p < 0.001$, with moderate effect size (Cohen's d = -0.52, -0.82 - -0.22 95% CI) and the difference between means of

PLOS Global Public Health

**Table 2. Data completeness and distribution of IPOS items at baseline assessment (n = 240).**

| Item | Frequency for each response score (%) | | | | | Mean (±SD) | Missing (%) |
|---|---|---|---|---|---|---|---|
| | Not at all (0) | Slight (1) | Moderate (2) | Severe (3) | Overwhelming/ all the time (4) | | |
| **Physical symptoms** | 0.4 | – | – | – | 0.0 | 14.0 (±6.4) | 10.8 |
| Pain | 14.2 | 17.9 | 29.6 | 22.9 | 15.4 | 2.1 (±1.3) | 0.0 |
| Shortness of breath | 59.1 | 14.8 | 12.7 | 8.9 | 4.6 | 0.9 (±1.2) | 1.3 |
| Weakness/lack of energy | 9.0 | 15.0 | 24.4 | 34.6 | 17.1 | 2.4 (±1.2) | 2.5 |
| Nausea* | 56.5 | 15.1 | 17.6 | 8.4 | 2.5 | 0.9 (±1.1) | 0.4 |
| Vomiting* | 66.5 | 14.6 | 11.3 | 6.3 | 1.3 | 0.6 (±1.0) | 0.4 |
| Poor appetite | 23.1 | 19.3 | 26.5 | 23.5 | 7.6 | 1.7 (±1.3) | 0.8 |
| Constipation* | 37.2 | 19.2 | 17.5 | 18.4 | 7.7 | 1.4 (±1.4) | 2.5 |
| Sore dry mouth | 43.6 | 16.9 | 21.2 | 11.9 | 6.4 | 1.2 (±1.3) | 1.7 |
| Drowsiness | 37.1 | 25.7 | 20.3 | 13.9 | 3.0 | 1.2 (±1.2) | 1.3 |
| Poor mobility | 30.4 | 15.4 | 12.9 | 21.7 | 19.6 | 1.9 (±1.5) | 0.0 |
| **Emotional symptoms** | 1.7 | – | – | – | 0.0 | 8.6 (±3.7) | 0.8 |
| Patient anxiety | 16.7 | 7.1 | 26.7 | 30.8 | 18.8 | 2.3 (±1.3) | 0.0 |
| Family anxiety | 14.2 | 7.5 | 25.1 | 41.8 | 11.3 | 2.3 (±1.2) | 0.4 |
| Depression | 17.9 | 13.8 | 28.3 | 27.5 | 12.5 | 2.0 (±1.3) | 0.0 |
| Feeling at peace | 10.0 | 19.7 | 43.5 | 15.9 | 10.9 | 2.0 (±1.1) | 0.4 |
| **Communication and Support** | 1.7 | – | – | – | 0.0 | 4.0 (±2.0) | 2.5 |
| Sharing feelings | 24.2 | 36.0 | 20.3 | 12.3 | 7.2 | 1.4 (±1.2) | 1.7 |
| Information | 31.2 | 50.4 | 5.4 | 7.5 | 5.4 | 1.1 (±1.1) | 0.0 |
| Practical problems | 7.1 | 41.2 | 43.7 | 6.7 | 1.3 | 1.5 (±0.8) | 0.8 |

*The mean of these items makes up the Gastrointestinal parcel score.

Physical subscale scores between advanced and early cancer patients is more than 0.5 standard deviations. Emotional and Communication and Support subscales did not discriminate well between patients with different disease severities.

*Responsiveness and Minimally Important Difference (MID):* We examined the ability of IPOS Hindi to detect changes in patients status [38] to inform sample size decisions evaluating effectiveness in future trials. We hypothesized that changes in IPOS Hindi scores would capture improvement or deterioration or would stay the same when there is no change in patient status based on patient-rated anchors. Self-rated anchor patients who showed *improvement* and responded that they were 'better' compared to the previous assessment were included in effect size calculations. Small to moderate effect sizes were observed in most subscales (Table D and E in S1 File). Standardised Responsive Mean (SRM) shows IPOS Hindi has small effect sizes for change from baseline to second follow-up assessment (Table F in S1 File).

A change of -1 for Physical and Communication & Support subscale, and -2 for emotional subscales suggests clinically important improvement; while a change of 1.8 in Physical, 1 in Emotional and 0.5 in Communication suggests clinically important deterioration in this sample (Table 4).

## Discussion

Standardized Patient-Reported outcome measures primarily use closed-ended items to support consistent scoring and psychometric evaluation. The open-ended questions of IPOS are unique, add depth to the standardized measurement data, and capture concerns that are critical, personal to the patient and their life, and direct care and consultation to address these key concerns. This part of IPOS is therefore valuable to guide holistic practice. This part could also provide further evidence of the content validity of the measure. Pain in the study sample was reported as the main concern, and

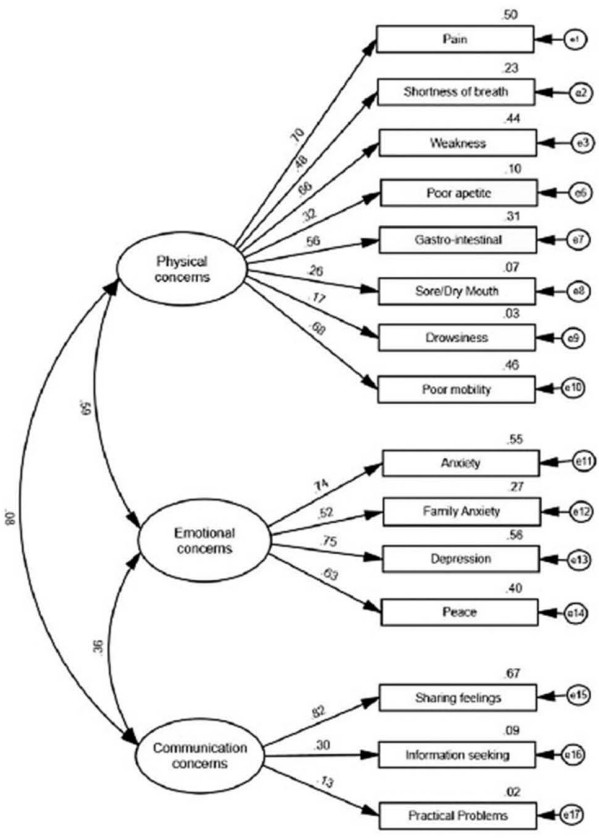

**Fig 2. CFA Model following post-hoc modifications (n = 240).**

this has also been observed in other studies with IPOS [9,42–45]. This could be because only a very small portion of the population who needs pain medications, has access to it in India [46,47]. However, morphine and other opiates critical to pain are highly regulated and even unavailable in other countries as a result of government bans [48], leading to what the World Health Organisation calls 'a preventable pain pandemic' [49] which impacts life greatly. Fatigue was also frequently reported as one of the main concerns. Previous studies have shown how the two symptoms, i.e., pain and fatigue are associated, and contribute significantly to cancer related suffering [50]. Study participants also reported struggling with financial difficulties, and how these disrupt their relationships and worry them over how their care would be funded.

**Table 3. Corrected Item-Total Correlations.**

| Subscale | Item | Corrected Item-Total Correlation |
|---|---|---|
| *Physical (n = 214)* | Pain | 0.40 |
| | Shortness of breath | 0.39 |
| | Weakness/lack of energy | 0.51 |
| | Poor appetite | 0.26 |
| | Sore dry mouth | 0.24 |
| | Drowsiness | 0.19 |
| | Poor mobility | 0.48 |
| | Gastro-Intestinal Parcel | 0.44 |
| *Emotional (n = 238)* | Patient anxiety | 0.61 |
| | Family anxiety | 0.43 |
| | Depression | 0.61 |
| | Feeling at peace | 0.47 |
| *Communication and Support (n = 234)* | Sharing feelings | 0.19 |
| | Information | 0.28 |
| | Practical problems | 0.14 |

IPOS yielded completeness of data with acceptable level of missingness. If missingness were to be high in future studies, we recommend considering appropriate imputation approaches; however, during initial validation studies, most authors caution against imputation, as it may distort psychometric properties [51]. Confirmatory factor analysis shows that IPOS Hindi demonstrates good fit with the hypothesized 3 factor model - Physical, Emotional and Communication & Support [12]. Twelve items demonstrated floor effects (≥15%), which may have limited sensitivity at the lower end of the scale. For this reason use of subscale scores to support clinical assessments rather than individual items is recommended. Similar, item floor effects have been observed in other studies with IPOS [52]. It is thus suggested that item-response theory-based approaches and re-calibration of items could be considered and explored in future work to improve measurement performance. The observed floor effects suggest that robust calibration will require larger, population-based samples [53]. Our current study was underpowered for this, but we plan to address it in future population-based work.

Overall IPOS Hindi has acceptable levels of reliability. Internal consistency, i.e., reliability of the Physical and Emotional subscales is moderate to high; however, the Communication and Support subscale is low. one of the reasons for this could be that this subscale contains only 3 items and that average correlations between the 3 items are low in this sample. The internal consistency of the third subscale, i.e., communication and support subscale was relatively low in our sample (Cronbach's α = 0.35). Notably, the original validation study also reported modest reliability for this subscale [54] (α = 0.58), suggesting that this dimension of the scale may have inherent limitations in internal consistency. Possible explanations include the small number of items, the heterogeneous nature of the construct, and contextual differences in how participants interpreted the items. While this indicates that findings related to this subscale should be interpreted with caution, the content remains theoretically important. Future work should consider revising or expanding the communication and support subscale by applying alternative reliability metrics, e.g., McDonald's omega, test–retest and item expansion to better capture the construct in diverse settings. Low engagement with the 'information' and 'sharing feelings' items may also reflect cultural norms, literacy barriers, or trust dynamics between participants and interviewers rather than lack of relevance. These factors warrant deeper investigation in future research. IPOS Physical and Emotional domains have demonstrated higher internal consistency in other settings and populations too [55,56]. Further studies could inform internal consistency. IPOS produces similar results over time in persons who have self-rated themselves as feeling the same. IPOS Hindi also shows good inter-rater reliability and with low measurement error.

**Table 4. Minimally Important Difference (MID) based on median change scores in patients who have improved or deteriorated from baseline to first follow-up assessment.**

|  | Subscale | Median Change Score | SD | n |
|---|---|---|---|---|
| **Improved** | Physical | -1 | 5.1 | 35 |
|  | Emotional | -2 | 3.6 | 43 |
|  | Communication & Support | -1 | 2.6 | 41 |
| **Deteriorated** | Physical | 1.8 | 7.7 | 20 |
|  | Emotional | 1.0 | 1.8 | 23 |
|  | Communication & Support | 0.5 | 2.3 | 22 |

IPOS Hindi has robust structural validity, where the associations between the items and subscales of IPOS and EQ-5D-5L are significant and in the predicted direction. These associations also reveal that in this population physical subscales correlate moderately with mobility and anxiety, and less so with pain. Physical subscales specifically show that patients living with advanced cancer experience more overwhelming effects of physical symptoms and have higher concerns and problems. The emotional burden seems to be similar irrespective of the disease stage in cancer patients. As Temoshok observed, the psychosocial impact could be related to the expected outcome at the point of diagnosis or later in disease progression [57].

IPOS Hindi is responsive to patient-anchored and reported changes, where small and moderate effect sizes are observed. A change of 1 and 2 respectively suggests improvement in Physical and Communication & Quality subscales. Further work is needed to establish MID with the emotional subscale. Other studies similarly have shown greater responsiveness of physical aspects, health status, and quality of life [58].

The study has several limitations. The three-factor model required parceling of GI items to achieve acceptable fit. Without parceling, fit indices were poor, indicating that this subdomain was unstable in our sample. While this justifies the analytic choice as used in other IPOS studies too [54], it alters the original IPOS construct and should be interpreted with caution. Future studies with larger samples should formally compare parceled and non-parcelled models to assess the robustness of the factor structure. Due to COVID-19 restrictions, data collection was restricted to Delhi NCR. Patients could not be recruited from other nearby states such as Uttar Pradesh, Haryana and Punjab in India, therefore this may affect the generalizability of our findings to Hindi Speaking populations in other regions. Ten patients had virtual second follow-up assessment rather than in-person interviews due to visiting restrictions and lockdowns in several localities. Our sample comprised only patients with cancer, and therefore the applicability of the Hindi IPOS to other chronic illnesses (such as organ failure or neurodegenerative conditions) remains to be established. Future validation in other Hindi-speaking regions and non-cancer populations is essential.

The qualitative component in our study was based on brief open-text comments, which limited the depth of qualitative analysis. While we presented word cloud visualizations for descriptive purposes, it offers limited analytic depth. Future work using more in-depth qualitative methods (e.g., interviews, focus groups) will be important to generate richer thematic insights and support item refinement. Future studies may benefit from thematic analysis with illustrative quotes, potentially triangulated with qualitative interviews, to strengthen interpretive depth. The Emotional and Communication subscales showed weak responsiveness and did not discriminate by disease stage, which raises concerns about their utility in longitudinal or intervention-based studies. Our estimates of Minimally Important Difference (MID) were based on small subgroups (n = 20–43), which constrain precision. These findings should be considered preliminary, and future studies with larger samples should confirm them using bootstrapped confidence intervals and anchor-based validation.

One of the key strengths of this study is that it is a critical collaborative study between two academic and research institutions and cancer charity, that engaged patients with lived experiences of socio-economic deprivation. Also, to the best of our knowledge, IPOS is the first validated tool to assess palliative outcomes in Hindi language.

## Conclusions

IPOS Hindi is a valid and reliable tool for assessing palliative care concerns among patients with cancer receiving home-based care in India. The Physical and Emotional subscales demonstrated acceptable validity and reliability and were sensitive to clinically important changes over time. Open-text responses provided additional insight into patients' needs.

Further work is needed to address floor effects and improve the Communication and Quality of Care subscale. Future studies should also examine IPOS Hindi in acute care settings and broader populations.

Aligned with global calls for better tools to measure suffering and guide palliative care implementation, IPOS Hindi offers a brief, person-centred measure suitable for use in low-resource settings.

## Supporting information

**S1 File. Supplementary Online File.**
(DOCX)

**S2 File. COSMIN Checklist.**
(DOC)

**S1 Checklist. Inclusivity in global research.**
(DOCX)

## Acknowledgments

We would like to acknowledge the support extended by CanSupport counsellors namely Surender Manjhi, Shashi Bhandari, Simranjeet, Sunita Sharma, Anju Sharma, Meena Sharma, Laxmi, Rajni Rajput, Jyoti Sharma, Shailly Sharma, Anjana Bisht, Anamika, Nikki, Poonam, Mausumi Bansali, Charanjeet Kaur, Divya Sharma, Rinky Pal, Anil Sharma, Ashish Samuel, and Renu Rajput for timely completion of data collection for the project. We also thank CanSupport Counselling Supervisors, Ms. Pallika Chaudhary and Mr. Nariender Gautam for monitoring field data collection.

IJH is a National Institute for Health Research (NIHR) Emeritus Senior Investigator and is supported by the NIHR Applied Research Collaboration (ARC) South London (SL) at King's College Hospital National Health Service Foundation Trust. IJH leads the Palliative and End of Life Care theme of the NIHR ARC SL and co-leads the national theme in this. RLC is funded by Cicely Saunders International and Marie Curie. MBH is supported by the NIHR ARC SL. The views expressed in this article are those of the authors and not necessarily those of the NIHR, or the Department of Health and Social Care.

**Research ethics and patient consent:** Ethics approval was received from the Institutional Ethics committee of CanSupport, a charity located in Delhi,India where the study was implemented with approval dated 6th December 2017. All patients provided written informed consent.

## Author contributions

**Conceptualization:** Tushti Bhardwaj, Irene J Higginson, Mevhibe B Hocaoglu.

**Data curation:** Tushti Bhardwaj.

**Formal analysis:** Tushti Bhardwaj, Rachel L Chambers, Harry Watson, Irene J Higginson, Mevhibe B Hocaoglu.

**Investigation:** Tushti Bhardwaj.

**Methodology:** Tushti Bhardwaj, Rachel L Chambers, Harry Watson, Ambika Rajvanshi, Reena Sharma, Irene J Higginson, Mevhibe B Hocaoglu.

**Project administration:** Tushti Bhardwaj.

**Resources:** Tushti Bhardwaj.

**Software:** Tushti Bhardwaj, Mevhibe B Hocaoglu.

**Supervision:** Tushti Bhardwaj, Ambika Rajvanshi, Reena Sharma, Irene J Higginson, Mevhibe B Hocaoglu.

**Validation:** Tushti Bhardwaj.

**Visualization:** Tushti Bhardwaj.

**Writing – original draft:** Tushti Bhardwaj, Rachel L Chambers, Harry Watson, Irene J Higginson, Mevhibe B Hocaoglu.

**Writing – review & editing:** Tushti Bhardwaj, Rachel L Chambers, Harry Watson, Ambika Rajvanshi, Reena Sharma, Irene J Higginson, Mevhibe B Hocaoglu.

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
