## [Decision Letter · Decision Letter 0]

15 Aug 2025

PGPH-D-25-01564

Evaluating the measurement properties of the Hindi IPOS in advanced cancer patients receiving home-based palliative care in India

Dear Dr. Bhardwaj,

Thank you for submitting your manuscript to PLOS Global Public Health. After careful consideration, we feel that it has merit but does not fully meet PLOS Global Public Health’s publication criteria as it currently stands. Therefore, we invite you to submit a revised version of the manuscript that addresses the points raised during the review process.

Please note that we have only been able to secure a single reviewer to assess your manuscript; the reviewers' comments can be found below. We are issuing a decision on your manuscript at this point to prevent further delays in the evaluation of your manuscript. Please be aware that the editor who handles your revised manuscript might find it necessary to invite additional reviewers to assess this work once the revised manuscript is submitted. However, we will aim to proceed on the basis of this single review if possible.

Could you please carefully revise the manuscript to address all comments raised?

We look forward to receiving your revised manuscript.

Kind regards,

Steve Zimmerman, PhD

PLOS Staff Editor

Journal Requirements:

- https://doi.org/10.1186/s12955-023-02102-4

In your revision ensure you cite all your sources (including your own works), and quote or rephrase any duplicated text outside the methods section. Further consideration is dependent on these concerns being addressed.

Additional Editor Comments (if provided):

Reviewers' comments:

Reviewer's Responses to Questions

**Comments to the Author**

1. Does this manuscript meet PLOS Global Public Health’s publication criteria? Is the manuscript technically sound, and do the data support the conclusions? The manuscript must describe methodologically and ethically rigorous research with conclusions that are appropriately drawn based on the data presented.

Reviewer #1: Yes

2. Has the statistical analysis been performed appropriately and rigorously?

Reviewer #1: Yes

3. Have the authors made all data underlying the findings in their manuscript fully available (please refer to the Data Availability Statement at the start of the manuscript PDF file)?

Reviewer #1: No

4. Is the manuscript presented in an intelligible fashion and written in standard English?

Reviewer #1: Yes

5. Review Comments to the Author

Reviewer #1: The study addresses a critical gap by validating a culturally adapted tool for Hindi-speaking patients.

Major Limitations and Suggestions

The internal consistency (Cronbach’s alpha = 0.35) is well below acceptable thresholds. Item-total correlations show weak discrimination, particularly for "Practical Problems" (0.14). Consider revising or expanding this domain for future iterations.

Twelve items show significant floor effects (≥15%), limiting sensitivity to change in less symptomatic patients. Authors should consider item response theory-based recalibration or alternate scoring techniques.

Pairwise deletion risks bias, and exclusion of subscale scores with single missing items limits clinical applicability. The authors might explore multiple imputation or prorated scoring options.

The three-factor structure required parceling of GI symptoms to achieve acceptable CFA fit indices. While justifiable, this alters the original IPOS construct and weakens cross-cultural comparability. Explicit justification and sensitivity analysis are encouraged.

The COVID-19-related restriction to Delhi NCR reduces generalizability to other Hindi-speaking regions (e.g., UP, Bihar). This should be more clearly stated in the limitations.

The sample includes only cancer patients. Validation for other chronic illnesses (e.g., organ failure, neurodegenerative diseases) is necessary to extend applicability of Hindi IPOS. In case that is not possible, the same should be explicitly mentioned.

Word cloud visualization offers limited insight into rich psychosocial concerns expressed in responses. A thematic analysis with illustrative quotes could enhance content validity and support item refinement.

Low engagement with "information" and "sharing feelings" items may reflect cultural norms or literacy barriers, or lack of trust between interviewer and responder. These require deeper investigation in future studies.

Emotional and Communication subscales showed weak responsiveness and no discriminative validity by disease stage. This questions their use in longitudinal or intervention-based studies.

Minimally Important Difference (MID) estimates are based on small subgroups (n=20–43), constraining precision. Bootstrapped confidence intervals or anchor-based validation in larger samples may be required.

Minor Issues

Typographical inconsistencies (e.g., “preset” study instead of “present”) should be corrected.

Figures (CFA model, word cloud) lack clear legends and numeric labels; revise for readability.

Data availability constraints could be elaborated to clarify ethical restrictions vs. operational barriers.

6. PLOS authors have the option to publish the peer review history of their article (what does this mean?). If published, this will include your full peer review and any attached files.

**Do you want your identity to be public for this peer review?** For information about this choice, including consent withdrawal, please see our Privacy Policy.

Reviewer #1: **Yes:**arun ghoshal

---

## [Decision Letter · Decision Letter 1]

10 Dec 2025

PGPH-D-25-01564R1

Evaluating the measurement properties of the Hindi IPOS in advanced cancer patients receiving home-based palliative care in India

Dear Dr. Bhardwaj,

Thank you for submitting your manuscript to PLOS Global Public Health. After careful consideration, we feel that it has merit but does not fully meet PLOS Global Public Health’s publication criteria as it currently stands. Therefore, we invite you to submit a revised version of the manuscript that addresses the points raised during the review process.

The manuscript has been evaluated by three reviewers, and their comments are available below. While Reviewers 1 and 2 express positive opinions of the study following the first round of revisions, the third Reviewer has raised concerns particularly related to the reporting.

Could you please revise the manuscript to carefully address the concerns raised?

We look forward to receiving your revised manuscript.

Kind regards,

Alejandro Torrado Pacheco, PhD

Staff Editor

Journal Requirements:

Additional Editor Comments (if provided):

Reviewers' comments:

Reviewer's Responses to Questions

**Comments to the Author**

1. If the authors have adequately addressed your comments raised in a previous round of review and you feel that this manuscript is now acceptable for publication, you may indicate that here to bypass the “Comments to the Author” section, enter your conflict of interest statement in the “Confidential to Editor” section, and submit your "Accept" recommendation.

Reviewer #1: All comments have been addressed

Reviewer #2: All comments have been addressed

Reviewer #3: (No Response)

2. Does this manuscript meet PLOS Global Public Health’s publication criteria? Is the manuscript technically sound, and do the data support the conclusions? The manuscript must describe methodologically and ethically rigorous research with conclusions that are appropriately drawn based on the data presented.

Reviewer #1: Yes

Reviewer #2: Yes

Reviewer #3: No

3. Has the statistical analysis been performed appropriately and rigorously?

Reviewer #1: Yes

Reviewer #2: Yes

Reviewer #3: Yes

4. Have the authors made all data underlying the findings in their manuscript fully available (please refer to the Data Availability Statement at the start of the manuscript PDF file)?

Reviewer #1: Yes

Reviewer #2: (No Response)

Reviewer #3: Yes

5. Is the manuscript presented in an intelligible fashion and written in standard English?

Reviewer #1: Yes

Reviewer #2: Yes

Reviewer #3: Yes

6. Review Comments to the Author

Reviewer #1: well done, few minor points can be acknowledged:

Communication & Support Subscale:

Internal consistency remains low (α = 0.35).

Future iterations should consider item expansion or alternative reliability metrics (e.g., McDonald’s omega).

Floor Effects:

Twelve items show ≥15% floor effects.

Consider item response theory (IRT) or recalibration in future studies.

Generalizability:

Sample restricted to Delhi NCR due to COVID-19.

Future validation in other Hindi-speaking regions and non-cancer populations is essential.

Qualitative Depth:

Word cloud visualization is informative but limited.

Future studies should incorporate thematic analysis with illustrative quotes.

Figures and Formatting:

Ensure all figures (e.g., CFA model, word cloud) have clear legends and numeric labels.

Minor typographical errors have been corrected, but a final proofread is recommended.

Data Sharing Statement:

Ethical and operational constraints are now clearly articulated.

Consider including a data access protocol for transparency.

Reviewer #2: This is a valuable study addressing the urgent need for validated, culturally appropriate outcome measures in India. Validating the IPOS in Hindi for home-based palliative care settings makes a significant contribution to the global public health and palliative care infrastructure in LMICs. Seems like my concerns have already been addressed in response to comments to previous reviewers. Respectfully

Reviewer #3: Title- Evaluating the measurement properties of the Hindi IPOS in advanced cancer patients receiving home-based palliative care in India- Abbreviations should be defined when mentioned for the first time to ensure clarity for readers who may not be familiar with the term.

Abstract

Methods-

1. The manuscript states that “This validation study was conducted with 240 adults living with cancer and receiving home-based palliative care in the Delhi National Capital Region of India.” Given that Delhi/NCR is a highly populated region and the national capital, the sample size of 240 patients appears relatively small compared to the total home-based palliative care population. It would strengthen the methodology section if the authors clarify from which specific centre(s) / home-based palliative care programme(s) these patients were recruited.

2. EQ-5D-5L - Abbreviations should be defined when mentioned for the first time to ensure clarity for readers who may not be familiar with the term.

Results

The statement regarding construct and known-groups validity being “supported by significant associations with EQ-5D-5L and cancer stage” remains incomplete without the presentation of statistical evidence. Please include the corresponding p-values in the results section to substantiate this claim.

Please mention the secondary objective too

Methodology

1. Please clarify whether all participant information was collected telephonically. If so, this should be stated clearly in the methodology section of the abstract, as the mode of data collection is an essential methodological detail.

2. Clarity regarding patient recruitment. Please indicate the number of eligible patients during the study duration, describe how they were approached, and report the final number enrolled. This information is essential for readers to evaluate the sampling process and overall robustness of the study design.

3. Please mention the inclusion and exclusion criteria of the participants

Results

1. The term “senior secondary (14 years of education)” is ambiguous without specifying the corresponding school standard. Please clearly state which class/grade this refers to (e.g., Class 12)

2. Socioeconomic status is referenced in the results, but the methodology does not describe how it was defined or assessed.

3. The presence of Stage I and II patients in a home-based palliative care cohort seems inconsistent with standard palliative care referral patterns, which predominantly include advanced-stage cases. Please clarify the staging assessment method and the referral criteria that resulted in the inclusion of early-stage patients.

Discussion

1. The authors justify the use of closed-ended questions by stating that such formats facilitate comparisons across groups. Since this is a single-group study, this rationale does not align with the methodology.

7. PLOS authors have the option to publish the peer review history of their article (what does this mean?). If published, this will include your full peer review and any attached files.

**Do you want your identity to be public for this peer review?** For information about this choice, including consent withdrawal, please see our Privacy Policy.

Reviewer #1: **Yes:**arun ghoshal

Reviewer #2: No

Reviewer #3: No

Figure Resubmissions:

---

## [Decision Letter · Decision Letter 2]

11 Mar 2026

Evaluating the measurement properties of the Hindi Integrated Palliative Care Outcome Scale (IPOS) in advanced cancer patients receiving home-based palliative care in India

PGPH-D-25-01564R2

Dear Dr Bhardwaj,

We are pleased to inform you that your manuscript 'Evaluating the measurement properties of the Hindi Integrated Palliative Care Outcome Scale (IPOS) in advanced cancer patients receiving home-based palliative care in India' has been provisionally accepted for publication in PLOS Global Public Health.

Best regards,

Sonali Sarkar

Academic Editor

Reviewer Comments (if any, and for reference):

Reviewer's Responses to Questions

**Comments to the Author**

1. If the authors have adequately addressed your comments raised in a previous round of review and you feel that this manuscript is now acceptable for publication, you may indicate that here to bypass the “Comments to the Author” section, enter your conflict of interest statement in the “Confidential to Editor” section, and submit your "Accept" recommendation.

Reviewer #1: All comments have been addressed

Reviewer #4: All comments have been addressed

Reviewer #5: All comments have been addressed

2. Does this manuscript meet PLOS Global Public Health’s publication criteria? Is the manuscript technically sound, and do the data support the conclusions? The manuscript must describe methodologically and ethically rigorous research with conclusions that are appropriately drawn based on the data presented.

Reviewer #1: Yes

Reviewer #4: Yes

Reviewer #5: Yes

3. Has the statistical analysis been performed appropriately and rigorously?

Reviewer #1: Yes

Reviewer #4: Yes

Reviewer #5: Yes

4. Have the authors made all data underlying the findings in their manuscript fully available (please refer to the Data Availability Statement at the start of the manuscript PDF file)?

Reviewer #1: Yes

Reviewer #4: Yes

Reviewer #5: Yes

5. Is the manuscript presented in an intelligible fashion and written in standard English?

Reviewer #1: Yes

Reviewer #4: Yes

Reviewer #5: Yes

6. Review Comments to the Author

Reviewer #1: (No Response)

Reviewer #4: Wek=ll drafted with robust analysis

Reviewer #5: Well written article.

7. PLOS authors have the option to publish the peer review history of their article (what does this mean?). If published, this will include your full peer review and any attached files.

**Do you want your identity to be public for this peer review?** For information about this choice, including consent withdrawal, please see our Privacy Policy.

Reviewer #1: **Yes:**arun ghoshal

Reviewer #4: No

Reviewer #5: No
